# Predtron: A Family of Online Algorithms for General Prediction Problems

**Prateek Jain**
Microsoft Research, INDIA
prajain@microsoft.com

**Nagarajan Natarajan**
University of Texas at Austin, USA
naga86@cs.utexas.edu

**Ambuj Tewari**
University of Michigan, Ann Arbor, USA
tewaria@umich.edu

## Abstract

Modern prediction problems arising in multilabel learning and learning to rank pose unique challenges to the classical theory of supervised learning. These problems have large prediction and label spaces of a combinatorial nature and involve sophisticated loss functions. We offer a general framework to derive mistake driven online algorithms and associated loss bounds. The key ingredients in our framework are a general loss function, a general vector space representation of predictions, and a notion of margin with respect to a general norm. Our general algorithm, Predtron, yields the perceptron algorithm and its variants when instantiated on classic problems such as binary classification, multiclass classification, ordinal regression, and multilabel classification. For multilabel ranking and subset ranking, we derive novel algorithms, notions of margins, and loss bounds. A simulation study confirms the behavior predicted by our bounds and demonstrates the flexibility of the design choices in our framework.

## 1   Introduction

Classical supervised learning problems, such as binary and multiclass classification, share a number of characteristics. The prediction space (the space in which the learner makes predictions) is often the same as the label space (the space from which the learner receives supervision). Because directly learning discrete valued prediction functions is hard, one learns real-valued or vector-valued functions. These functions generate continuous predictions that are converted into discrete ones via simple mappings, e.g., via the 'sign' function (binary classification) or the 'argmax' function (multiclass classification). Also, the most commonly used loss function is simple, viz. the 0-1 loss.

In contrast, modern prediction problems, such as multilabel learning, multilabel ranking, and subset ranking do not share these characteristics. In order to handle these problems, we need a more general framework that offers more flexibility. First, it should allow for the possibility of having different label space and prediction space. Second, it should allow practitioners to use creative, new ways to map continuous, vector-valued predictions to discrete ones. Third, it should permit the use of general loss functions.

Extensions of the theory of classical supervised learning to modern predictions problems have begun. For example, the work on calibration dimension [1] can be viewed as extending one aspect of the theory, viz. that of calibrated surrogates and consistent algorithms based on convex optimization. This paper deals with the extension of another interesting part of classical supervised learning: mistake driven algorithms such as perceptron (resp. winnow) and their analyses in terms of $\ell_2$ (resp. $\ell_1$) margins [2, Section 7.3].

We make a number of contributions. First, we provide a general framework (Section 2) whose ingredients include an arbitrary loss function and an arbitrary representation of discrete predictions in a continuous space. The framework is abstract enough to be of general applicability but it offers enough mathematical structure so that we can derive a general online algorithm, Predtron (Algorithm 1), along with an associated loss bound (Theorem 1) under an abstract margin condition (Section 2.2). Second, we show that our framework unifies several perception-like algorithms for classical problems such as binary classification, multiclass classification, ordinal regression, and multilabel classification (Section 3). Even for these classical problems, we get some new results, for example, when the loss function treats labels asymmetrically or when there exists a 'reject' option in classification. Third, we apply our framework to two modern prediction problems: subset ranking (Section 4) and multilabel ranking (Section 5). In both of these problems, the prediction space (rankings) is different from the supervision space (set of labels or vector of relevance scores). For these two problems, we propose interesting, novel notions of correct prediction with a margin and derive mistake bounds under a loss derived from NDCG, a ranking measure that pays more attention to the performance at the top of a ranked list. Fourth, our techniques based on online convex optimization (OCO) can effortlessly incorporate notions of margins w.r.t. non-Euclidean norms, such as $\ell_1$ norm, group norm, and trace norm. Such flexibility is important in modern prediction problems where the learned parameter can be a high dimensional vector or a large matrix with low group or trace norm. Finally, we test our theory in a simulation study (Section 6) dealing with the subset ranking problem showing how our framework can be adapted to a specific prediction problem. We investigate different margin notions as we vary two key design choices in our abstract framework: the map used to convert continuous predictions into discrete ones, and the choice of the norm used in the definition of margin.

**Related Work.** Our general algorithm is related to the perceptron and online gradient descent algorithms used in structured prediction [3, 4]. But, to the best of knowledge, our emphasis on keeping label and prediction spaces possibly distinct, our use of a general representation of predictions, and our investigation of generalized notions of margins are all novel. The use of simplex coding in multiclass problems [5] inspired the use of maximum similarity/minimum distance decoding to obtain discrete predictions from continuous ones. Our proofs use results about Online Gradient Descent and Online Mirror Descent from the Online Convex Optimization literature [6].

## 2 Framework and Main Result

The key ingredients in classic supervised learning are an input space, an output space and a loss function. In this paper, the input space $\mathcal{X} \in \mathbb{R}^p$ will always be some subset of a finite dimensional Euclidean space. Our algorithms maintain prediction functions as a linear combination of the seen inputs. As a result, they easily kernelize and the theory extends, in a straightforward way to the case when the input space is a, possibly infinite dimensional, reproducing kernel Hilbert space (RKHS).

### 2.1 Labels, Prediction, and Scores

We will distinguish between the label space and the prediction space. The former is the space where the training labels come from whereas the latter is the space where the learning algorithm has to make predictions in. Both spaces will be assumed to be finite. Therefore, without any loss of generality, we can identify the label space with $[\ell] = \{1, \ldots, \ell\}$ and the prediction space with $[k]$ where $\ell, k$ are positive, but perhaps very large, integers. A given loss function $\mathbf{L} : [k] \times [\ell] \to \mathbb{R}_+$ maps a prediction $\sigma \in [k]$ and a label $y \in [\ell]$ to a non-negative loss $\mathbf{L}(\sigma, y)$. The loss $\mathbf{L}$ can equivalently be thought of as a $k \times \ell$ matrix with loss values as entries. Define the set of correct predictions for a label $y$ as $\Sigma_y = \{\sigma_y \in [k] : \mathbf{L}(\sigma_y, y) = 0\}$. We assume that, for every label $y$, the set $\Sigma_y$ is non-empty. That is, every column of the loss matrix has a zero entry. Also, let $c_\mathbf{L} = \min_{\mathbf{L}(\sigma,y)>0} \mathbf{L}(\sigma, y)$ and $C_\mathbf{L} = \max_{\sigma,y} \mathbf{L}(\sigma, y)$ be the minimum (non-zero) and maximum entries in the loss matrix.

In an online setting, the learner will see a stream of examples $(X_\tau, Y_\tau) \in \mathcal{X} \times [\ell]$. Learner will predict scores using a linear predictor $W \in \mathbb{R}^{d \times p}$. However, the predicted scores $W X_\tau$ will be in $\mathbb{R}^d$, not in the prediction space $[k]$. So, we need a function $\mathrm{pred} : \mathbb{R}^d \to [k]$ to convert scores into actual predictions. We will assume that there is a unique representation $\mathrm{rep}(\sigma) \in \mathbb{R}^d$ of each

prediction $\sigma$ such that $\|\operatorname{rep}(\sigma)\|_2 = 1$ for all $\sigma$. Given this, a natural transformation of scores into prediction is given by the following *maximum similarity decoding*:

$$\operatorname{pred}(t) \in \underset{\sigma \in [k]}{\operatorname{argmax}} \; \langle \operatorname{rep}(\sigma), t \rangle \; , \tag{1}$$

where ties in the "argmax" can be broken arbitrarily. There are some nice consequences of the definition of $\operatorname{pred}$ above. First, because $\|\operatorname{rep}(\sigma)\|_2 = 1$, maximum similarity decoding is equivalent to *nearest neighbor decoding*: $\operatorname{pred}(t) \in \operatorname{argmin}_\sigma \|\operatorname{rep}(\sigma) - t\|_2$. Second, we have a homogeneity property: $\operatorname{pred}(ct) = \operatorname{pred}(t)$ if $c > 0$. Third, $\operatorname{rep}$ serves as an "inverse" of $\operatorname{pred}$ in the following sense. We have, $\operatorname{pred}(\operatorname{rep}(\sigma)) = \sigma$ for all $\sigma$. Moreover, $\operatorname{rep}(\operatorname{pred}(t))$ is more similar to $t$ than the representation of any other prediction $\sigma$:

$$\forall t \in \mathbb{R}^d, \sigma \in [k], \; \langle \operatorname{rep}(\operatorname{pred}(t)), t \rangle \geq \langle \operatorname{rep}(\sigma), t \rangle \, .$$

In view of these facts, we will use $\operatorname{pred}^{-1}(\sigma)$ and $\operatorname{rep}(\sigma)$ interchangeably. Using $\operatorname{pred}$, the loss function $L$ can be extended to a function defined on $\mathbb{R}^d \times [k]$ as:

$$\mathbf{L}(t, y) = \mathbf{L}(\operatorname{pred}(t), y).$$

With a little abuse of notation, we will continue to denote this new function also by $\mathbf{L}$.

## 2.2 Margins

We say that *a score $t$ is compatible with a label $y$* if the set of $\sigma$'s that achieve the maximum in the definition (1) of $\operatorname{pred}$ is exactly $\Sigma_y$. That is, $\operatorname{argmax}_{\sigma \in [k]} \; \langle \operatorname{pred}^{-1}(\sigma), t \rangle = \Sigma_y$. Hence, for any $\sigma_y \in \Sigma_y, \sigma \notin \Sigma_y$, we have $\langle \operatorname{pred}^{-1}(\sigma_y), t \rangle > \langle \operatorname{pred}^{-1}(\sigma), t \rangle$. The notion of margin makes this requirement stronger. We say that a score $t$ *has a margin $\gamma > 0$ on label $y$*, iff $t$ is compatible with $y$ and

$$\forall \sigma_y \in \Sigma_y, \sigma \notin \Sigma_y, \; \langle \operatorname{pred}^{-1}(\sigma_y), t \rangle \geq \langle \operatorname{pred}^{-1}(\sigma), t \rangle + \gamma$$

Note that margin scales with $t$: if $t$ has margin $\gamma$ on $y$ then $ct$ has margin $c\gamma$ on $y$ for any positive $c$. If we are using linear predictions $t = WX$, we say that *$W$ has margin $\gamma$ on $(X, y)$* iff $t = WX$ has margin $\gamma$ on $y$. We say that *$W$ has margin $\gamma$ on a dataset $(X_1, y_1), \dots, (X_n, y_n)$* iff $W$ has margin $\gamma$ on $(X_\tau, y_\tau)$ for all $\tau \in [n]$. Finally, a dataset $(X_1, y_1), \dots, (X_n, y_n)$ is said to be *linearly separable with margin $\gamma$* if there is a unit norm[1] $W^\star$ such that $W^\star$ has margin $\gamma$ on $(X_1, y_1), \dots, (X_n, y_n)$.

## 2.3 Algorithm

Just like the classic perceptron algorithm, our generalized perceptron algorithm (Algorithm 1) is mistake driven. That is, it only updates on round when a mistake, i.e., a non-zero loss, is incurred. On a mistake round, it makes a rank-one update of the form $W_{\tau+1} = W_\tau - g_\tau \cdot X_\tau^\top$ where $g_\tau \in \mathbb{R}^d, X_\tau \in \mathbb{R}^p$. Therefore, $W_\tau$ always has a representation of the form $\sum_i g_i X_i^\top$. The prediction on a fresh input $X$ is given by $\sum_i g_i \langle X_i, X \rangle$ which means the algorithm, just like the original perceptron, can be kernelized.

We will give a loss bound for the algorithm using tools from Online Convex Optimization (OCO). Define the function $\phi : \mathbb{R}^d \times [\ell] \to \mathbb{R}$ as

$$\phi(t, y) = \max_{\sigma \in [k]} \left( \mathbf{L}(\sigma, y) - \langle \operatorname{pred}^{-1}(\sigma_y) - \operatorname{pred}^{-1}(\sigma), t \rangle \right) \tag{2}$$

where $\sigma_y \in \Sigma_y$ is an arbitrary member of $\Sigma_y$. For any $y$, $\phi(\cdot, y)$ is a point-wise maximum of linear functions and hence convex. Also, $\phi$ is non-negative: choose $\sigma = \sigma_y$ to lower bound the maximum. The inner product part vanishes and the loss $\mathbf{L}(\sigma_y, y)$ vanishes too because $\sigma_y \in \Sigma_y$. Given the definition of $\phi$, Algorithm 1 can be described succinctly as follows. At round $\tau$, if $\mathbf{L}(W_\tau X_\tau, Y_\tau) > 0$, then $W_{\tau+1} = W_\tau - \eta \nabla_W \phi(W X_\tau, Y_\tau)$, otherwise $W_{\tau+1} = W_\tau$.

**Algorithm 1** Predtron: Extension of the Perceptron Algorithm to General Prediction Problems

---
1: $W_1 \leftarrow \mathbf{0}$
2: **for** $\tau = 1, 2, \ldots$ **do**
3:    Receive $X_\tau \in \mathbb{R}^p$
4:    Predict $\sigma_\tau = \mathrm{pred}(W_\tau X_\tau) \in [k]$
5:    Receive label $y_\tau \in [\ell]$
6:    **if** $\mathbf{L}(\sigma_\tau, y_\tau) > 0$ **then**
7:       $(t, y) = (W_\tau X_\tau, y_\tau)$
8:       $\tilde{\sigma}_\tau = \mathrm{argmax}_{\sigma \in [k]} \left( \mathbf{L}(\sigma, y) - \left\langle \mathrm{pred}^{-1}(\sigma_y) - \mathrm{pred}^{-1}(\sigma), t \right\rangle \right) \in [k]$
9:       $\nabla_\tau = (\mathrm{pred}^{-1}(\tilde{\sigma}_\tau) - \mathrm{pred}^{-1}(\sigma_y)) \cdot X_\tau^\top \in \mathbb{R}^{d \times p}$
10:       $W_{\tau+1} = W_\tau - \eta \nabla_\tau$
11:    **else**
12:       $W_{\tau+1} = W_\tau$
13:    **end if**
14: **end for**

---

**Theorem 1.** *Suppose the dataset* $(X_1, y_1), \ldots, (X_n, y_n)$ *is linearly separable with margin* $\gamma$. *Then the sequence* $W_\tau$ *generated by Algorithm 1 with* $\eta = c_{\mathbf{L}}/(4R^2)$ *satisfies the loss bound,*

$$\sum_{\tau=1}^n \mathbf{L}(W_\tau X_\tau, y_\tau) \leq \frac{4R^2 C_{\mathbf{L}}^2}{c_{\mathbf{L}} \gamma^2}$$

*where* $\|X_\tau\|_2 \leq R$ *for all* $\tau$.

Note that the bound above assumes perfect linear separability. However, just the classic perceptron, the bound will degrade gracefully when the best linear predictor does not have enough margin on the data set.

The Predtron algorithm has some interesting variants, two of which we consider in the appendix. A loss driven version, Predtron.LD, enjoys a loss bound that gets rid of the $C_{\mathbf{L}}/c_{\mathbf{L}}$ factor in the bound above. A version, Predtron.Link, that uses link functions to deal with margins defined with respect to non-Euclidean norms is also considered.

## 3 Relationship to Existing Results

It is useful to discuss a few concrete applications of the abstract framework introduced in the last section. Several existing loss bounds can be readily derived by applying our bound for the generalized perceptron algorithm in Theorem 1. In some cases, our framework yields a different algorithm than existing counterparts, yet admitting identical loss bounds, up to constants.

**Binary Classification.** We begin with the classical perceptron algorithm for binary classification (i.e., $\ell = 2$) [7]: $\mathbf{L}_{0\text{-}1}(\sigma, y) = 1$ if $\sigma \neq y$ or 0 otherwise. Letting $\mathrm{rep}(\sigma)$ be $+1$ for the positive class and $-1$ for the negative class, predictor vector $W_\tau \in \mathbb{R}^{1 \times p}$, and thus $\mathrm{pred}(t) = \mathrm{sign}(t)$, Algorithm 1 reduces to the original perceptron algorithm; Theorem 1 yields identical mistake bound on a linearly separable dataset with margin $\gamma$ (if the classical margin is $\gamma$, ours works out to be $2\gamma$), i.e. $\sum_{\tau=1}^n \mathbf{L}_{0\text{-}1}(W_\tau X_\tau, y_\tau) \leq \frac{R^2}{\gamma^2}$. We can also easily incorporate asymmetric losses. Let $\mathbf{L}_\alpha(\sigma, y) = \alpha_y$, if $\sigma \neq y$ and 0 otherwise. We then have the following result.

**Corollary 2.** *Consider the perceptron with weighted loss* $\mathbf{L}_\alpha$. *Assume* $\alpha_1 \geq \alpha_2$ *without loss of generality. Then the sequence* $W_\tau$ *generated by Algorithm 1 satisfies the weighted mistake bound,*

$$\sum_{\tau=1}^n \mathbf{L}_\alpha(W_\tau X_\tau, y_\tau) \leq \frac{4R^2 \alpha_1^2}{\alpha_2^2 \gamma^2}.$$

We are not aware of such results for weighted loss. Previous work [8] studies perceptrons with uneven margins, and the loss bound there only implies a bound on the unweighted loss: $\sum_{\tau=1}^n \mathbf{L}_{0\text{-}1}(t_\tau, y_\tau)$. In a technical note, Rätsch and Kivinen [9] provide a mistake bound of the

form (without proof): $\sum_{\tau=1}^{n} \mathbf{L}_\alpha(W_\tau X_\tau, y_\tau) \leq \frac{R^2}{4\gamma^2}$, but for the specific choice of weights $\alpha_1 = a^2$ and $\alpha_2 = (1-a)^2$ for any $a \in [0,1]$.

Another interesting extension is obtained by allowing the predictions to have a REJECT option. Define $\mathbf{L}_{\text{REJ}}(\text{REJECT}, y) = \beta_y$ and $\mathbf{L}_{\text{REJ}}(\sigma, y) = \mathbf{L}_{0\text{-}1}(\sigma, y)$ otherwise. Assume $1 \geq \beta_1 \geq \beta_2 > 0$ without loss of generality. Choosing the standard basis vectors in $\mathbb{R}^2$ to be $\text{rep}(\sigma)$ for the positive and the negative classes, and $\text{rep}(\text{REJECT}) = \frac{1}{\sqrt{2}} \sum_{\sigma \in \{1,2\}} \text{rep}(\sigma)$, we obtain $\sum_{\tau=1}^{n} \mathbf{L}_{\text{REJ}}(W_\tau X_\tau, y_\tau) \leq \frac{4R^2\beta_1^2}{\gamma^2\beta_2^2}$ (See Appendix C.1).

**Multiclass Classification.** Each instance is assigned exactly one of $m$ classes (i.e., $\ell = m$). Extending binary classification, we choose the standard basis vectors in $\mathbb{R}^m$ to be $\text{rep}(\sigma)$ for the $m$ classes. The learner predicts score $t \in \mathbb{R}^m$ using the predictor $W \in \mathbb{R}^{m \times p}$. So, $\text{pred}(t) = \arg\max_i t_i$. Let $w_j$ denote the $j$th row of $W$ (corresponding to label $j$). The definition of margin becomes:
$$\langle w_y, X \rangle - \max_{j \neq y} \langle w_j, X \rangle \geq \gamma$$

which is identical to the multiclass margin studied earlier [10]. For the multiclass 0-1 loss $\mathbf{L}_{0\text{-}1}$, we recover their bound, up to constants[2]. Moreover, our surrogate $\phi$ for $\mathbf{L}_{0\text{-}1}$:

$$\phi(t, y) = \max\big(0, 1 + \max_{\sigma \neq y} t_\sigma - t_y\big),$$

matches the multiclass extension of the Hinge loss studied by [11]. Finally, note that it is straightforward to obtain loss bounds for multiclass perceptron with REJECT option by naturally extending the definitions of $\text{rep}$ and $\mathbf{L}_{\text{REJ}}$ for the binary case.

**Ordinal Regression.** The goal is to assign ordinal classes (such as ratings) to a set of objects $\{X_1, X_2, \dots\}$ described by their features $X_i \in \mathbb{R}^p$. In many cases, precise rating information may not be available, but only their relative ranks; i.e., the observations consist of object-rank pairs $(X_\tau, y_\tau)$ where $y_\tau \in [\ell]$. $\mathcal{Y}$ is totally-ordered with ">" relation, which in turn induces a partial ordering on the objects ($X_j$ is preferred to $X_{j'}$ if $y_j > y_{j'}$, $X_j$ and $X_{j'}$ are not comparable if $y_j = y_{j'}$). For the ranking loss $\mathbf{L}(\sigma, y) = |\sigma - y|$, the PRank perceptron algorithm [12] enjoys the bound $\sum_{\tau=1}^{n} \mathbf{L}(\tau_\tau, y_\tau) \leq (\ell - 1)(R^2 + 1)/\tilde{\gamma}^2$, where $\tilde{\gamma}$ is a certain rank margin. By a reduction to multi-class classification with $\ell$ classes, Algorithm 1 achieves the loss bound $4(\ell - 1)^2 R^2 / \gamma^2$ (albeit, for a different margin $\gamma$).

**Multilabel Classification.** This setting generalizes multiclass classification in that instances are assigned subsets of $m$ classes rather than unique classes, i.e., $\ell = 2^m$. The loss function $\mathbf{L}$ of interest may dictate the choice of $\text{rep}$ and in turn $\text{pred}$. For example, consider the following subset losses that treat labels as well as predictions as subsets: (i) Subset 0-1 loss: $\mathbf{L}_{\text{IsErr}}(\sigma, y) = 1$ if $\sigma = y$ or 0 otherwise; (ii) Hamming loss: $\mathbf{L}_{\text{Ham}}(\sigma, y) = |\sigma \cup y| - |\sigma \cap y|$, and (ii) Error set size: $\mathbf{L}_{\text{ErrSetSize}}(\sigma, y) = \big|\{(r, s) \in y \times ([m] \setminus y) : r \notin \sigma, s \in \sigma\}\big|$. A natural choice of $\text{rep}$ then is the subset indicator vector in $\{+1, -1\}^d$, where $d = m = \log \ell$, which can be expressed as $\text{rep}(\sigma) = \frac{1}{\sqrt{m}}\big(\sum_{j \in \sigma} e_j - \sum_{j \notin \sigma} e_j\big)$ (where $e_j$'s are the standard basis vectors in $\mathbb{R}^m$). The learner predicts score $t \in \mathbb{R}^m$ using a matrix $W \in \mathbb{R}^{m \times p}$. Note that $\text{pred}(t) = \text{sign}(t)$, where sign is applied component-wise. The number of predictions is $2^m$, but we show in Appendix C.2 that the surrogate (2) and its gradient can be efficiently computed for all of the above losses.

## 4  Subset Ranking

In subset ranking [13], the task is to learn to rank a number of documents in order of their relevance to a query. We will assume, for simplicity, that the number of documents per query is constant that we denote by $m$. The input space is a subset of $\mathbb{R}^{m \times p_0}$ that we can identify with $\mathbb{R}^p$ for $p = mp_0$. Each row of an input matrix corresponds to a $p_0$-dimensional feature vector derived jointly using the query

and one of the documents associated with it. The predictions $\sigma$ are all $m!$ permutations of degree $m$. The most natural (but by no means the only one) representation of permutations is to set $\text{rep}(\sigma) = -\sigma/Z$ where $\sigma(i)$ is the position of the document $i$ in the predicted ranking and the normalization $Z$ ensures that $\text{rep}(\sigma)$ is a unit vector. Note that the dimension $d$ of this representation is equal to $m$. The minus sign in this representation ensures that $\text{pred}(t)$ outputs a permutation that corresponds to sorting the entries of $t$ in *decreasing* order, a common convention in existing work. A more general representation is obtained by setting $\text{rep}(\sigma) = f(\sigma)/Z$ where $f : \mathbb{R} \to \mathbb{R}$ is a strictly decreasing real valued function that is applied entry-wise to $\sigma$. The normalization $Z = \sqrt{\sum_{i=1}^m f^2(i)}$ ensures that $\|\text{rep}(\sigma)\|_2 = 1$. To convert an input matrix $X \in \mathbb{R}^p$ ($p = mp_0$) into a score vector $t \in \mathbb{R}^m$, it seems that we need to learn a matrix $W \in \mathbb{R}^{m \times mp_0}$. However, a natural permutation invariance requirement (if the documents associated are presented in a permuted fashion, the output scores should also get permuted in the same way) reduces the dimensionality of $W$ to $p_0$ (see, e.g., [14] for more details). Thus, given a vector $w \in \mathbb{R}^{p_0}$ we get the score vector as $t = Xw$. The label space consists of relevance score vectors $y \in \{0, 1, \ldots, Y_{\max}\}^m$ where $Y_{\max}$ is typically between $1$ and $4$ (yielding $2$ to $5$ grades of relevance). Note that the prediction space (of size $k = m!$) is different from the label space (of size $\ell = (Y_{\max} + 1)^m$).

A variety of loss functions have been used in subset ranking. For multigraded relevance judgments, a very popular choice is NDCG which is defined as $NDCG(\sigma, y) = \left( \sum_{i=1}^m \frac{2^{y(i)-1}}{\log_2(1+\sigma(i))} \right)/Z(y)$ where $Z(y)$ is a normalization constant ensuring NDCG stays bounded by 1. To convert it into a loss we define $\mathbf{L}_{\text{NDCG}} = 1 - NDCG$. Note that any permutation that sorts $y$ in decreasing order gets zero $\mathbf{L}_{\text{NDCG}}$. One might worry that the computation of the surrogate defined in (2) and its gradient might require an enumeration of $m!$ permutations. The next lemma allays such a concern.

**Lemma 3.** *When $\mathbf{L} = \mathbf{L}_{\text{NDCG}}$ and $\text{rep}(\sigma)$ is chosen as above, the computation of the surrogate (2), as well as its gradient, can be reduced to solving a linear assignment problem and hence can be done in $O(m^3)$ time.*

We now give a result explaining what it means for a score vector $t$ to have a margin $\gamma$ on $y$ when we use a representation of the form described above. Without loss of generality, we may assume that $y$ is sorted in decreasing order of relevance judgements.

**Lemma 4.** *Suppose $\text{rep}(\sigma) = f(\sigma)/Z$ for a strictly decreasing function $f : \mathbb{R} \to \mathbb{R}$ and $Z = \sqrt{\sum_{i=1}^m f^2(i)}$. Let $y$ be a non-constant relevance judgement vector sorted in decreasing order. Suppose $i_1 < i_2, \ldots < i_N, N \geq 1$ are the positions where the relevance drops by a grade or more (i.e., $y(i_j) < y(i_j - 1)$). Then $t$ has a margin $\gamma$ on $y$ iff $t$ is compatible with $y$ and, for $j \in [N]$,*

$$t_{i_j - 1} \geq t_{i_j} + \frac{\gamma Z}{f(i_j - 1) - f(i_j)}$$

*where we define $i_0 = 1, i_{N+1} = m + 1$ to handle boundary cases.*

Note that if we choose $f(i) = -i^\alpha, \alpha > 1$ then $f(i_j - 1) - f(i_j) = O(i_j^{\alpha-1})$ for large $i_j$. In that case, the margin condition above requires less separation between documents with different relevance scores down the list (when viewed in decreasing order of relevance scores) than at the top of the list. We end this section with a loss bound for $\mathbf{L}_{\text{NDCG}}$ under a margin condition.

**Corollary 5.** *Suppose $\mathbf{L} = \mathbf{L}_{\text{NDCG}}$ and $\text{rep}(\sigma)$ is as in Lemma 4. Then, assuming the dataset is linearly separable with margin $\gamma$, the sequence generated by Algorithm 1 with line 9 replaced by*

$$\nabla_\tau = X_\tau^\top (\text{pred}^{-1}(\tilde{\sigma}_\tau) - \text{pred}^{-1}(\sigma_y)) \in \mathbb{R}^{p_0 \times 1}$$

*satisfies*

$$\sum_{\tau=1}^n \mathbf{L}_{\text{NDCG}}(X_\tau w_\tau, y_\tau) \leq \frac{2^{Y_{\max}+3} \cdot m^2 \log_2^2(2m) \cdot R^2}{\gamma^2}$$

*where $\|X_\tau\|_{\text{op}} \leq R$.*

Note that the result above uses the standard $\ell_2$-norm based notion of margin. Imagine a subset ranking problem, where only a small number of features are relevant. It is therefore natural to consider a notion of margin where the weight vector that ranks everything perfectly has low group $\ell_1$ norm, instead of low $\ell_2$ norm. The $\ell_1$ margin also appears in the analysis of AdaBoost [2, Definition

6.2]. We can use a special case of a more general algorithm given in the appendix (Appendix B.2, Algorithm 3). Specifically, we replace line 10 with the step $w_{\tau+1} = (\nabla\psi)^{-1}(\nabla\psi(w_\tau) - \nabla_\tau)$ where $\psi(w) = \frac{1}{2}\|w\|_r^2$. We set $r = \log(p_0)/(\log(p_0) - 1)$. The mapping $\nabla\psi$ and its inverse can both be easily computed (see, e.g., [6, p. 145]).

**Corollary 6.** *Suppose* $\mathbf{L} = \mathbf{L}_{\mathrm{NDCG}}$ *and* $\mathrm{rep}(\sigma)$ *is as in Lemma 4. Then, assuming the dataset is linearly separable with margin* $\gamma$ *by a unit* $\ell_1$ *norm* $w^\star$ *(*$\|w^\star\|_1 = 1$*), the sequence generated by Algorithm 3 with* $\psi$ *chosen as above (and line 9 modified as in Corollary 5), satisfies*

$$\sum_{\tau=1}^{n} \mathbf{L}_{\mathrm{NDCG}}(X_\tau w_\tau, y_\tau) \leq \frac{9 \cdot 2^{Y_{\max}+3} \cdot m^2 \log_2^2(2m) \cdot R^2 \cdot \log p_0}{\gamma^2}$$

*where* $\max_{j=1,\ldots,p_o} \|X_{\tau,j}\|_2 \leq R$ *and* $X_{\tau,j}$ *denotes the jth column of* $X_\tau$.

## 5 Multilabel Ranking

As discussed in Section 3, in multilabel *classification*, both prediction space and label space are $\{0,1\}^m$ with sizes $k = \ell = 2^m$. In multilabel *ranking*, however, the learner has to output rankings as predictions. So, as in the previous section, we have $k = m!$ since the prediction $\sigma$ can be any one of $m!$ permutations of the labels. As before, we choose $\mathrm{rep}(\sigma) = f(\sigma)/Z$ and hence $d = m$. However, unlike the previous section, the input is no longer a matrix but a vector $X \in \mathbb{R}^p$. A prediction $t \in \mathbb{R}^d$ is obtained as $WX$ where $W \in \mathbb{R}^{m \times p}$. Note the contrast with the last section: there, inputs are matrices and a weight vector is learned; here, inputs are vectors and a weight matrix is learned. Since we output rankings, it is reasonable to use a loss that takes positions of labels into account. We can use $\mathbf{L} = \mathbf{L}_{\mathrm{NDCG}}$. Algorithm 1 now immediately applies. Lemma 3 already showed that is efficiently implementable. We have the following straightforward corollary.

**Corollary 7.** *Suppose* $\mathbf{L} = \mathbf{L}_{\mathrm{NDCG}}$ *and* $\mathrm{rep}(\sigma)$ *is as in Lemma 4. Then, assuming the dataset is linearly separable with margin* $\gamma$*, the sequence generated by Algorithm 1 satisfies*

$$\sum_{\tau=1}^{n} \mathbf{L}_{\mathrm{NDCG}}(X_\tau w_\tau, y_\tau) \leq \frac{2^{Y_{\max}+3} \cdot m^2 \log_2^2(2m) \cdot R^2}{\gamma^2}$$

*where* $\|X_\tau\|_2 \leq R$.

The bound above matches the corresponding bound, up to loss specific constants, for the multiclass multilabel perceptron (MMP) algorithm studied by [15]. The definition of margin by [15] for MMP is different from ours since their algorithms are designed *specifically for multilabel ranking*. Just like them, we can also consider other losses, e.g., precision at top $K$ positions. Another perceptron style algorithm for multilabel ranking adopts a pairwise approach of comparing two labels at a time [16]. However, no loss bounds are derived.

The result above uses the standard Frobenius norm based margin. Imagine a multilabel problem, where only a small number of features are relevant across all labels. Then, it is natural to consider a notion of margin where the matrix that ranks everything perfectly has low group $(2,1)$ norm, instead of low Frobenius norm, where $\|W\|_{2,1} = \sum_{j=1}^p \|W_j\|_2$ ($W_j$ denotes a column of $W$). We again use a special case of Algorithm 3 (Appendix B.2). Specifically, we replace line 10 with the step $W_{\tau+1} = (\nabla\psi)^{-1}(\nabla\psi(W_\tau) - \nabla_\tau)$ where $\psi(W) = \frac{1}{2}\|W\|_{2,r}^2$. Recall that the group $(2,r)$-norm is the $\ell_r$ norm of the $\ell_2$ norm of the columns of $W$. We set $r = \log(p)/(\log(p) - 1)$. The mapping $\nabla\psi$ and its inverse can both be easily computed (see, e.g., [17, Eq. (2)]).

**Corollary 8.** *Suppose* $\mathbf{L} = \mathbf{L}_{\mathrm{NDCG}}$ *and* $\mathrm{rep}(\sigma)$ *is as in Lemma 4. Then, assuming the dataset is linearly separable with margin* $\gamma$ *by a unit group norm* $W^\star$ *(*$\|W^\star\|_{2,1} = 1$*), the sequence generated by Algorithm 3 with* $\psi$ *chosen as above, satisfies*

$$\sum_{\tau=1}^{n} \mathbf{L}_{\mathrm{NDCG}}(X_\tau w_\tau, y_\tau) \leq \frac{9 \cdot 2^{Y_{\max}+3} \cdot m^2 \log_2^2(2m) \cdot R^2 \cdot \log p}{\gamma^2}$$

*where* $\|X_\tau\|_\infty \leq R$.

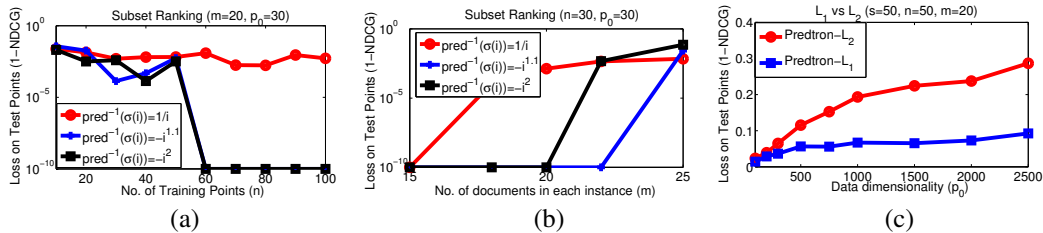

Figure 1: Subset Ranking: NDCG loss for different $\text{pred}^{-1}$ choices with varying $n$ (Plot (a)) and $m$ (Plot (b)). As predicted by Lemmas 4 and 5, $\text{pred}^{-1}(\sigma_i) = -i^{1.1}$ is more accurate than $1/i$. (c): $L_1$ vs $L_2$ margin. $\mathbf{L}_{\text{NDCG}}$ for two different Predtron algorithms based on $L_1$ and $L_2$ margin. Data is generated using $L_1$ margin notion but with varying sparsity of the optimal scoring function $w^*$.

## 6    Experiments

We now present simulation results to demonstrate the application of our proposed Predtron framework to subset ranking. We also demonstrate that empirical results match the trend predicted by our error bounds, hence hinting at tightness of our (upper) bounds. Due to lack of space, we focus only on the subset ranking problem. Also, we would like to stress that we do not claim that the basic version of Predtron itself (with $\eta = 1$) provides a state-of-the-art ranker. Instead, we wish to demonstrate the applicability and flexibility of our framework in a controlled setting.

We generated $n$ data points $X_\tau \in \mathbb{R}^{m \times p_0}$ using a Gaussian distribution with independent rows. The $i$th row of $X_\tau$ represents a document and is sampled from a spherical Gaussian centered at $\mu_i$. We selected a $w^* \in \mathbb{R}^{p_0}$ and also a set of thresholds $[\zeta_1, \ldots, \zeta_{m+1}]$ to generate relevance scores; we set $\zeta_j = \frac{1}{j}, \forall 2 \leq j \leq m$ and $\zeta_1 = +\infty$ and $\zeta_{m+1} = -\infty$. We set relevance score $y_\tau(i)$ of the $i$th document in the $\tau$th document-set as: $y_\tau(i) = m - j$ iff $\zeta_{j+1} \leq \langle X_\tau(i), w^* \rangle \leq \zeta_j$. That is, $y_\tau(i) \in [m-1]$.

We measure performance of a given method using the NDCG loss $\mathbf{L}_{\text{NDCG}}$ defined in Section 4. Note that $\mathbf{L}_{\text{NDCG}}$ is less sensitive to errors in predictions for the less relevant documents in the list. On the other hand, our selection of thresholds $\zeta_i$'s implies that the *gap* between scores of lower-ranked documents is very small compared to the higher-ranked ones, and hence chances of making mistakes lower down the list is higher.

Figure 1 (a) shows $\mathbf{L}_{\text{NDCG}}$ (on a test set) for our Predtron algorithm (see Section 4) but with different $\text{pred}^{-1}$ functions. For $\text{pred}^{-1}(\sigma(i)) = f_2(\sigma) = -i^{1.1}$, $f_2(i-1) - f_2(i)$ is monotonically increasing with $i$. On the other hand, for $\text{pred}^{-1}(\sigma(i)) = f_1(\sigma) = 1/i$, $f_1(i-1) - f_1(i)$ is monotonically decreasing with $i$. Lemma 4 shows that the mistake bound (in terms of $\mathbf{L}_{\text{NDCG}}$) of Predtron is better when $\text{pred}^{-1}$ function is selected to be $f_2(\sigma(i)) = -i^{1.1}$ (as well as for $f_3(\sigma(i)) = -i^2$) instead of $f_1(\sigma(i)) = 1/i$. Clearly, Figure 1 (a) empirically validates this mistake bound with $\mathbf{L}_{\text{NDCG}}$ going to almost 0 for $f_2$ and $f_3$ with just 60 training points, while $f_1$ based Predtron has large loss even with $n = 100$ training points.

Next, we fix the number of training instances to be $n = 30$ and vary the number of documents $m$. As the gap between $\zeta_i$'s decreases for larger $i$, increasing $m$ implies reducing the margin. Naturally, Predtron with the above mentioned inverse functions has monotonically increasing loss (see Figure 1 (b)). However, $f_2$ and $f_3$ provide zero-loss solutions for larger $m$ when compared to $f_1$.

Finally, we conduct an experiment to show that by selecting appropriate notion of margin, Predtron can obtain more accurate solutions. To this end, we generate data from $[-1, 1]^{p_0}$ and select a sparse $w^*$. Now, Predtron with $\ell_2$-margin notion, i.e., standard gradient descent has $\sqrt{p_0}$ dependency in the error bounds while the $\ell_1$-margin (see Corollary 6) has only $s \log(p_0)$ dependence. This error dependency is also revealed by Figure 1 (c), where increasing $p_0$ with fixed $s$ leads to minor increase in the loss for $\ell_1$-based Predtron but leads to significantly higher loss for $\ell_2$-based Predtron.

**Acknowledgments**

A. Tewari acknowledges the support of NSF under grant IIS-1319810.

## Footnotes

[1] Here, we mean that the Frobenius norm $\|W^\star\|_F$ equals 1. Of course, the notion of margin can be generalized to any norm including the entry-based $\ell_1$ norm $\|W\|_1$ and the spectrum-based $\ell_1$ norm $\|W\|_{S(1)}$ (also called the nuclear or trace norm). See Appendix B.2.

[2]Perceptron algorithm in [10] is based on a slightly different loss defined as $\mathbf{L}_{\text{ErrSet}}(t, y) = 1$ if $|\{r \neq y : t_r \geq t_y\}| > 0$ or 0 otherwise (where $t = WX$). This loss upper bounds $\mathbf{L}_{0\text{-}1}$ (because of the way ties are handled, there can be rounds when $\mathbf{L}_{0\text{-}1}$ is 0, but $\mathbf{L}_{\text{ErrSet}}$ is 1).

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
