[Supplementary Material]

# A Proof of Theorem 1

Before we prove the main result, we provide a couple of useful lemmas. The first shows that $\phi(t, y)$ is an upper bound on $\mathbf{L}(t, y)$.

**Lemma 9.** *For any $t \in \mathbb{R}^d, y \in [\ell]$, we have $\phi(t, y) \geq \mathbf{L}(t, y)$.*

*Proof.* We have,

$$\max_{\sigma \in [k]} \left( \mathbf{L}(\sigma, y) - \langle \mathrm{pred}^{-1}(\sigma_y) - \mathrm{pred}^{-1}(\sigma), t \rangle \right)$$
$$\geq \mathbf{L}(\mathrm{pred}(t), y) - \langle \mathrm{pred}^{-1}(\sigma_y) - \mathrm{pred}^{-1}(\mathrm{pred}(t)), t \rangle$$
$$\geq \mathbf{L}(t, y).$$

Note that the last step is by the similarity maximization property of $\mathrm{pred}^{-1}(\mathrm{pred}(t))$. □

The next lemma proves a key self-bounding property of the derivative of $\phi(t, y)$, w.r.t. $t$, that is crucial for the analysis of the generalized perceptron algorithm to go through.

**Lemma 10.** *Fix a $t \in \mathbb{R}^d, y \in [\ell]$ such that $\mathbf{L}(t, y) > 0$. Then, we have*

$$\|\nabla_t \phi(t, y)\|_2^2 \leq \frac{4}{c_{\mathbf{L}}} \mathbf{L}(t, y)$$

*Proof.* Note that $\nabla_t \phi(t, y)$ is

$$\mathrm{pred}^{-1}(\tilde{\sigma}_{t,y}) - \mathrm{pred}^{-1}(\sigma_y)$$

where

$$\tilde{\sigma}_{t,y} = \operatorname*{argmax}_{\sigma \in [k]} \left( \mathbf{L}(\sigma, y) - \langle \mathrm{pred}^{-1}(\sigma_y) - \mathrm{pred}^{-1}(\sigma), t \rangle \right)$$

So, $\|\nabla_t \phi(t, y)\|_2^2 \leq (\| \mathrm{pred}^{-1}(\tilde{\sigma}_{t,y})\|_2 + \| \mathrm{pred}^{-1}(\sigma_y)\|_2)^2 \leq 4$. On the other hand, on a mistake round, $\mathbf{L}(t, y) \geq c_{\mathbf{L}}$. □

Finally, the lemma below states that if $t$ gets large enough margin on a label $y$ then $\phi(t, y)$ is zero.

**Lemma 11.** *If $t \in \mathbb{R}^d$ has margin $\gamma \geq C_{\mathbf{L}}$ on $y$, then $\phi(t, y) = 0$.*

*Proof.* Note that for any $\sigma \notin \Sigma_y$,

$$\mathbf{L}(\sigma, y) - \langle \mathrm{pred}^{-1}(\sigma_y) - \mathrm{pred}^{-1}(\sigma), t \rangle \leq \mathbf{L}(\sigma, y) - \gamma \leq \mathbf{L}(\sigma, y) - C_{\mathbf{L}} \leq 0$$

For any $\sigma'_y \in \Sigma_y$, we have $\mathbf{L}(\sigma'_y, y) = 0$ and

$$\langle \mathrm{pred}^{-1}(\sigma_y), t \rangle = \langle \mathrm{pred}^{-1}(\sigma'_y), t \rangle = \max_{\sigma} \langle \mathrm{pred}^{-1}(\sigma), t \rangle.$$

Therefore, $\phi(t, y) = 0$. □

Now we have all the ingredients to prove the main result.

*Proof of Theorem 1.* Recall that a mistake round is one where $\mathbf{L}(W_\tau X_\tau, Y_\tau) > 0$. Define the following sequence of convex functions:

$$f_\tau(W) = \begin{cases} 0 & \text{on non-mistake round} \\ W \mapsto \phi(W X_\tau, Y_\tau) & \text{on mistake round} \end{cases}$$

Consider online gradient descent (OGD) updates: $W_{\tau+1} = W_\tau - \eta \nabla f_\tau(W_\tau)$. Standard OGD analysis (see, e.g., [6, Eq. (2.15)]) implies that, for any $W$ (we will deal with the issue of choosing $\eta$ shortly):

$$\sum_{\tau=1}^{n} f_\tau(W_\tau) \leq \sum_{\tau=1}^{n} f_\tau(W) + \frac{\eta}{2} \sum_{\tau=1}^{n} \|\nabla_\tau\|_F^2 + \frac{\|W\|_F^2}{2\eta} \tag{3}$$

where $\nabla_\tau = \nabla_W f_\tau(W_\tau)$.

On non-mistake rounds, the gradient as well as loss, are both zero. On mistake rounds, the gradient is

$$\nabla_\tau = \nabla_W f_\tau(W_\tau) = \nabla_t \phi(W_\tau X_\tau, Y_\tau) X_\tau^\top$$

and therefore

$$\|\nabla_\tau\|_F^2 \leq \frac{4}{c_{\mathbf{L}}} R^2 \mathbf{L}(W_\tau X_\tau, Y_\tau)$$

by the self-bounding property (Lemma 10) and boundedness of $X_\tau$. Therefore, we have

$$\frac{\eta}{2} \sum_{\tau=1}^n \|\nabla_\tau\|_F^2 \leq \frac{2\eta R^2}{c_{\mathbf{L}}} \sum_{\tau=1}^n \mathbf{L}(W_\tau X_\tau, Y_\tau) \tag{4}$$

On non-mistake rounds, $f_\tau$ as well as loss, are both zero. On mistake rounds,

$$f_\tau(W_\tau) = \phi(W_\tau X_\tau, Y_\tau) \geq \mathbf{L}(W_\tau X_\tau, Y_\tau)$$

by upper bound property of $\phi$ (Lemma 9). So we also have

$$\sum_{\tau=1}^n \mathbf{L}(W_\tau X_\tau, Y_\tau) \leq \sum_{\tau=1}^n f_\tau(W_\tau) \tag{5}$$

Now plugging in (5) and (4) into (3), we get

$$\sum_{\tau=1}^n \mathbf{L}(W_\tau X_\tau, Y_\tau) \leq \sum_{\tau=1}^n f_\tau(W) + \frac{2\eta R^2}{c_{\mathbf{L}}} \sum_{\tau=1}^n \mathbf{L}(W_\tau X_\tau, Y_\tau) + \frac{\|W\|_F^2}{2\eta}$$

By assumption, the sequence $(X_\tau, y_\tau)$ is linearly separable with margin $\gamma$. That is, there exists a $W^\star$ with margin $\gamma$ on $(X_\tau, y_\tau)$. By the scaling property of margin, this means that $W = C_{\mathbf{L}} W^\star/\gamma$ has margin $C_{\mathbf{L}}$ on $(X_\tau, y_\tau)$. For this $W$, by Lemma 11, we have $\sum_{\tau=1}^n f_\tau(W) = 0$. Since $\|W\|_F^2 \leq C_{\mathbf{L}}^2/\gamma^2$, we have the bound

$$\left(1 - \frac{2\eta R^2}{c_{\mathbf{L}}}\right) \sum_{\tau=1}^n \mathbf{L}(W_\tau X_\tau, Y_\tau) \leq \frac{C_{\mathbf{L}}^2}{2\gamma^2 \eta}$$

and choosing $\eta = c_{\mathbf{L}}/(4R^2)$ gives the bound

$$\sum_{\tau=1}^n \mathbf{L}(W_\tau X_\tau, Y_\tau) \leq \frac{4R^2 C_{\mathbf{L}}^2}{c_{\mathbf{L}} \gamma^2}$$

$\square$

## B   Algorithm Variants

We provide two variants of Predtron. First, we present Predtron.LD, a loss driven version that uses a surrogate that is not dependent on the loss but incorporates the loss in the stepsize. Then, we present Predtron.Link, a version that allows for margin to be defined w.r.t. an arbitrary norm and uses an appropriate link function in its updates.

### B.1   Choice of Surrogate

Consider using the surrogate:

$$\phi_1(t, y) = \max\{0, 1 + \max_{\sigma \notin \Sigma_y} \langle \text{pred}^{-1}(\sigma), t \rangle - \langle \text{pred}^{-1}(\sigma_y), t \rangle\}$$

For any $y$, $\phi_1(t, y)$ is obviously non-negative and convex in $t$. Moreover, when a mistake is made this surrogate is at least 1.

**Lemma 12.** *Suppose $t, y$ are such that $\mathbf{L}(t, y) > 0$. Then $\phi_1(t, y) \geq 1$.*

*Proof.* Since $\mathbf{L}(t, y) > 0$, there exists $\sigma \notin \Sigma_y$ such that $\langle \mathrm{pred}^{-1}(\sigma), t \rangle = \max_{\sigma'} \langle \mathrm{pred}^{-1}(\sigma'), t \rangle$. Therefore, we have

$$\max_{\sigma \notin \Sigma_y} \langle \mathrm{pred}^{-1}(\sigma), t \rangle \geq \langle \mathrm{pred}^{-1}(\sigma_y), t \rangle$$

and therefore $\phi_1(t, y) \geq 1$. $\qquad\square$

The surrogate $\phi_1$ is also zero given large enough margin.

**Lemma 13.** *If $t \in \mathbb{R}^d$ has margin $\gamma \geq 1$ on $y$, then $\phi_1(t, y) = 0$.*

*Proof.* Note that for any $\sigma \notin \Sigma_y$,

$$1 - \langle \mathrm{pred}^{-1}(\sigma_y) - \mathrm{pred}^{-1}(\sigma), t \rangle \leq 1 - \gamma \leq 0.$$

Therefore,

$$1 + \max_{\sigma \notin \Sigma_y} \langle \mathrm{pred}^{-1}(\sigma), t \rangle - \langle \mathrm{pred}^{-1}(\sigma_y), t \rangle \leq 0$$

and hence $\phi_1(t, y) = 0$. $\qquad\square$

---

**Algorithm 2** Predtron.LD: A Loss Driven Version of Predtron

1: $W_1 \leftarrow \mathbf{0}$
2: **for** $\tau = 1, 2, \dots$ **do**
3:     Receive $X_\tau \in \mathbb{R}^p$
4:     Predict $\sigma_\tau = \mathrm{pred}(W_\tau X_\tau) \in [k]$
5:     Receive label $y_\tau \in [\ell]$
6:     **if** $\mathbf{L}(\sigma_\tau, y_\tau) > 0$ **then**
7:         $(t, y) = (W_\tau X_\tau, y_\tau)$
8:         $\tilde{\sigma}_\tau = \mathrm{argmax}_{\sigma \notin \Sigma_y} \langle \mathrm{pred}^{-1}(\sigma), t \rangle \in [k]$
9:         $\nabla_\tau = (\mathrm{pred}^{-1}(\tilde{\sigma}_\tau) - \mathrm{pred}^{-1}(\sigma_y)) \cdot X_\tau^\top \in \mathbb{R}^{d \times p}$
10:      $W_{\tau+1} = W_\tau - \eta \, \mathbf{L}(\sigma_\tau, y_\tau) \cdot \nabla_\tau$
11:     **else**
12:      $W_{\tau+1} = W_\tau$
13:     **end if**
14: **end for**

---

**Theorem 14.** *Suppose the dataset $(X_1, y_1), \dots, (X_n, y_n)$ is linearly separable with margin $\gamma$. Then the sequence $W_\tau$ generated by Algorithm 2 with $\eta = 1/(4C_{\mathbf{L}} R^2)$ satisfies the loss bound*

$$\sum_{\tau=1}^{n} \mathbf{L}(W_\tau X_\tau, y_\tau) \leq \frac{4R^2 C_{\mathbf{L}}}{\gamma^2}$$

*where $\|X_\tau\|_2 \leq R$ for all $\tau$.*

*Proof.* As before, let a mistake round be one where $\mathbf{L}(W_\tau X_\tau, Y_\tau) > 0$. Let $\mathbf{L}_\tau = \mathbf{L}(W_\tau X_\tau, y_\tau)$. Define the following sequence of convex functions:

$$f_\tau(W) = \mathbf{L}_\tau \cdot \phi_1(W X_\tau, y_\tau).$$

Algorithm 2 is simply running online gradient descent (OGD) updates: $W_{\tau+1} = W_\tau - \nabla f_\tau(W_\tau)$. This is trivial to see for non-mistake round. On mistake rounds, observe that the outer maximum in the definition of $\phi_1$ is not achieved at 0 and hence the gradient is given by

$$\nabla_t \phi(t, y) = \mathrm{pred}^{-1}(\tilde{\sigma}) - \mathrm{pred}^{-1}(\sigma_y)$$

where

$$\tilde{\sigma} = \mathrm{argmax}_{\sigma \notin \Sigma_y} \langle \mathrm{pred}^{-1}(\sigma), t \rangle.$$

Setting $\nabla_\tau = \nabla_W f_\tau(W_\tau)$ and using standard OGD analysis (see, e.g., [6, Eq. (2.15)]) we get that, for any $W$:

$$
\begin{aligned}
\sum_{\tau=1}^{n} f_\tau(W_\tau) &\leq \sum_{\tau=1}^{n} f_\tau(W) + \frac{\eta}{2} \sum_{\tau=1}^{n} \|\nabla_\tau\|_F^2 + \frac{\|W\|_F^2}{2\eta} \\
&\leq \sum_{\tau=1}^{n} f_\tau(W) + \frac{\eta}{2} \sum_{\tau=1}^{n} \mathbf{L}_\tau^2 \cdot \|\nabla_t \phi_1(X_\tau W_\tau, y_\tau)\|_2^2 \cdot \|X_\tau\|_2^2 + \frac{\|W\|_F^2}{2\eta} \\
&\leq \sum_{\tau=1}^{n} f_\tau(W) + \frac{\eta}{2} \sum_{\tau=1}^{n} C_{\mathbf{L}} \cdot \mathbf{L}_\tau \cdot 4 \cdot R^2 + \frac{\|W\|_F^2}{2\eta} \\
&= \sum_{\tau=1}^{n} f_\tau(W) + 2\eta C_{\mathbf{L}} R^2 \sum_{\tau=1}^{n} \mathbf{L}_\tau + \frac{\|W\|_F^2}{2\eta}.
\end{aligned}
$$

By Lemma 12, we know that $\mathbf{L}_\tau \leq f_\tau(W_\tau)$. Further, by assumption, the sequence $(X_\tau, y_\tau)$ is linearly separable with margin $\gamma$. That is, there exists a $W^\star$ with margin $\gamma$ on $(X_\tau, y_\tau)$. By the scaling property of margin, this means that $W = W^\star/\gamma$ has margin 1 on $(X_\tau, y_\tau)$. For this $W$, by Lemma 13, we have $\sum_{\tau=1}^{n} f_\tau(W) = 0$. Since $\|W\|_F^2 \leq 1/\gamma^2$, we have the bound

$$
\left(1 - 2\eta C_{\mathbf{L}} R^2\right) \sum_{\tau=1}^{n} \mathbf{L}_\tau \leq \frac{1}{2\gamma^2 \eta}
$$

and choosing $\eta = 1/(4 C_{\mathbf{L}} R^2)$ gives the bound

$$
\sum_{\tau=1}^{n} \mathbf{L}(W_\tau X_\tau, Y_\tau) \leq \frac{4 R^2 C_{\mathbf{L}}}{\gamma^2}.
$$

$\square$

## B.2 Choice of Norm

---
**Algorithm 3** Predtron.Link: Predtron with a Link Function $(\nabla\psi)^{-1}$

---
1: $\Theta_1 \leftarrow \mathbf{0}; W_1 = (\nabla\psi)^{-1}(\Theta_1)$
2: **for** $\tau = 1, 2, \ldots$ **do**
3:     Receive $X_\tau \in \mathbb{R}^p$
4:     Predict $\sigma_\tau = \mathrm{pred}(W_\tau X_\tau) \in [k]$
5:     Receive label $y_\tau \in [\ell]$
6:     **if** $\mathbf{L}(\sigma_\tau, y_\tau) > 0$ **then**
7:         $(t, y) = (W_\tau X_\tau, y_\tau)$
8:         $\tilde{\sigma}_\tau = \mathrm{argmax}_{\sigma \in [k]} \left(\mathbf{L}(\sigma, y) - \langle \mathrm{pred}^{-1}(\sigma_y) - \mathrm{pred}^{-1}(\sigma), t \rangle\right) \in [k]$
9:         $\nabla_\tau = (\mathrm{pred}^{-1}(\tilde{\sigma}_\tau) - \mathrm{pred}^{-1}(\sigma_y)) \cdot X_\tau^\top \in \mathbb{R}^{d \times p}$
10:        $\Theta_{\tau+1} = \Theta_\tau - \eta \nabla_\tau; W_{\tau+1} = (\nabla\psi)^{-1}(\Theta_{\tau+1})$
11:     **else**
12:        $\Theta_{\tau+1} = \Theta_\tau; W_{\tau+1} = W_\tau$
13:     **end if**
14: **end for**

---

Let $\|\cdot\|$ be a norm and $\psi(W) = \frac{1}{2}\|W\|^2$ be $\alpha$-strongly convex w.r.t. $\|\cdot\|$. Note that $\psi(\mathbf{0}) = 0$ and $\nabla\psi(cW) = c\nabla\psi(W)$ for $c > 0$. Consider Algorithm 3, a version of Algorithm 1 that uses inverse of the mapping $\nabla\psi$ to generate iterates. Since $\psi$ is strongly convex, the mapping $\nabla\psi$ is indeed invertible. Let $\|\cdot\|_\star$ be the norm dual to $\|\cdot\|$. Since the gradients $\nabla_\tau$ are rank one, we need one additional property. Suppose there exists a norm $\|\|\cdot\|\|$ such that, for any $u \in \mathbb{R}^d, v \in \mathbb{R}^p$, we have

$$
\|uv^\top\|_\star \leq \|u\|_2 \cdot \|\|v\|\| \tag{6}
$$

We can now prove a loss bound for Algorithm 3.

**Theorem 15.** *Suppose* $\psi, \|\cdot\|, \|\cdot\|_\star, \|\!|\cdot|\!\|$ *are as above. In particular, let* $\psi$ *be* $\alpha$-*strongly convex w.r.t.* $\|\cdot\|$. *Suppose the dataset* $(X_1, y_1), \ldots, (X_n, y_n)$ *is linearly separable by a unit norm* $W^\star$ *(*$\|W^\star\| = 1$*), by margin* $\gamma$. *Then, Algorithm 3 with* $\eta = \alpha c_{\mathbf{L}}/(4R^2)$ *satisfies the loss bound*

$$\sum_{\tau=1}^{n} \mathbf{L}(W_\tau X_\tau, Y_\tau) \leq \frac{4R^2 C_{\mathbf{L}}^2}{\alpha c_{\mathbf{L}} \gamma^2}$$

*where* $\|\!|X_\tau|\!\| \leq R$.

*Proof.* Recall that a mistake round is one where $\mathbf{L}(W_\tau X_\tau, Y_\tau) > 0$. Define the following sequence of convex functions:

$$f_\tau(W) = \begin{cases} 0 & \text{on non-mistake round} \\ W \mapsto \phi(WX_\tau, Y_\tau) & \text{on mistake round} \end{cases}$$

Consider online mirror descent (OMD) updates: $\nabla\psi(W_{\tau+1}) = \nabla\psi(W_\tau) - \eta\nabla f_\tau(W_\tau)$. Standard OMD analysis (see, e.g., [6, Theorem 2.21]) implies that, for any $W$ (we will deal with the issue of choosing $\eta$ shortly):

$$\sum_{\tau=1}^{n} f_\tau(W_\tau) \leq \sum_{\tau=1}^{n} f_\tau(W) + \frac{\eta}{2\alpha}\sum_{\tau=1}^{n}\|\nabla_\tau\|_\star^2 + \frac{\|W\|^2}{2\eta} \tag{7}$$

where $\nabla_\tau = \nabla_W f_\tau(W_\tau)$.

On non-mistake rounds, the gradient as well as loss, are both zero. On mistake rounds, the gradient is

$$\nabla_\tau = \nabla_W f_\tau(W_\tau) = \nabla_t \phi(W_\tau X_\tau, Y_\tau) X_\tau^\top$$

and therefore

$$\|\nabla_\tau\|_\star^2 \leq \|\nabla_t \phi(W_\tau X_\tau, Y_\tau)\|_2^2 \cdot \|\!|X_\tau|\!\|^2 \leq \frac{4}{c_{\mathbf{L}}}R^2 \mathbf{L}(W_\tau X_\tau, Y_\tau)$$

by the inequality (6), the self-bounding property (Lemma 10), and boundedness of $X_\tau$. Therefore, we have

$$\frac{\eta}{2\alpha}\sum_{\tau=1}^{n}\|\nabla_\tau\|_F^2 \leq \frac{2\eta R^2}{\alpha c_{\mathbf{L}}}\sum_{\tau=1}^{n}\mathbf{L}(W_\tau X_\tau, Y_\tau) \tag{8}$$

On non-mistake rounds, $f_\tau$ as well as loss, are both zero. On mistake rounds,

$$f_\tau(W_\tau) = \phi(W_\tau X_\tau, Y_\tau) \geq \mathbf{L}(W_\tau X_\tau, Y_\tau)$$

by upper bound property of $\phi$ (Lemma 9). So we also have

$$\sum_{\tau=1}^{n}\mathbf{L}(W_\tau X_\tau, Y_\tau) \leq \sum_{\tau=1}^{n} f_\tau(W_\tau) \tag{9}$$

Now plugging in (9) and (8) into (7), we get

$$\sum_{\tau=1}^{n}\mathbf{L}(W_\tau X_\tau, Y_\tau) \leq \sum_{\tau=1}^{n} f_\tau(W) + \frac{2\eta R^2}{\alpha c_{\mathbf{L}}}\sum_{\tau=1}^{n}\mathbf{L}(W_\tau X_\tau, Y_\tau) + \frac{\|W\|^2}{2\eta}$$

By assumption, the sequence $(X_\tau, y_\tau)$ is linearly separable with margin $\gamma$. That is, there exists a $W^\star$ with margin $\gamma$ on $(X_\tau, y_\tau)$. By the scaling property of margin, this means that $W = C_{\mathbf{L}}W^\star/\gamma$ has margin $C_{\mathbf{L}}$ on $(X_\tau, y_\tau)$. For this $W$, by Lemma 11, we have $\sum_{\tau=1}^{n} f_\tau(W) = 0$. Since $\|W\|^2 \leq C_{\mathbf{L}}^2/\gamma^2$, we have the bound

$$\left(1 - \frac{2\eta R^2}{\alpha c_{\mathbf{L}}}\right)\sum_{\tau=1}^{n}\mathbf{L}(W_\tau X_\tau, Y_\tau) \leq \frac{C_{\mathbf{L}}^2}{2\gamma^2\eta}$$

and choosing $\eta = \alpha c_{\mathbf{L}}/(4R^2)$ gives the bound

$$\sum_{\tau=1}^{n}\mathbf{L}(W_\tau X_\tau, Y_\tau) \leq \frac{4R^2 C_{\mathbf{L}}^2}{\alpha c_{\mathbf{L}} \gamma^2}$$

$\square$

## C    Details of Results in Section 3

### C.1    Classification with REJECT option

We note that the separability requirement to allow REJECT option is more stringent than the standard classification in the following sense. If a dataset is linearly separable with margin $\gamma$ for the standard classification, it may no longer be linearly separable with the same margin $\gamma$ if we allow REJECT option. The other way, however, holds true. This is observed by examining the definition of margin requirement in Section 2.2. Consider $1 \geq \beta_1 \geq \beta_2 > 0$. Then, a score $t \in \mathbb{R}^2$ has a margin $\gamma > 0$ on label $y = 1$, iff $\operatorname{pred}(t) = 1$ and $t_1 \geq \max\left(t_2, \frac{t_1+t_2}{\sqrt{2}}\right) + \gamma$, and a margin $\gamma > 0$ on label $y = 2$, iff $\operatorname{pred}(t) = 2$ and $t_2 \geq \max\left(t_1, \frac{t_1+t_2}{\sqrt{2}}\right) + \gamma$. For the case $\beta_2 = 0$ (i.e. instance of class 2 can be predicted as REJECT without penalty), then a score $t \in \mathbb{R}^2$ has a margin $\gamma > 0$ on label $y = 2$, iff $\operatorname{pred}(t) \in \{2, \text{REJECT}\}$ and $t_1 \leq \min\left(t_2, \frac{t_1+t_2}{\sqrt{2}}\right) - \gamma$.

### C.2    Subset losses for Multilabel learning

For a given $y$ and $t$, using the definition of rep in Section 3 for multilabel learning, the surrogate (2) for a given loss $\mathbf{L}$ can be expressed as:

$$\phi(t, y) = -\left\langle \operatorname{pred}^{-1}(\sigma_y), t \right\rangle + \max_{\mathbf{v} \in \{+1, -1\}^m} \mathbf{L}(\mathbf{v}, y) + \langle \mathbf{v}, t \rangle.$$

Clearly, we can compute the surrogate (and its gradient) efficiently if we can compute the max efficiently. Define the indicator function $I(P) = 1$ if predicate $P$ is true or 0 otherwise. Let:

$$a = \sum_i I(v_i = 1)I(i \in y), \qquad\qquad b = \sum_i I(v_i = 1)I(i \notin y),$$

$$c = \sum_i I(v_i = -1)I(i \in y), \qquad\qquad d = \sum_i I(v_i = -1)I(i \notin y).$$

In the following, we show that the max in the surrogate can be computed in time $O(m^2)$, for any loss which can be expressed as a function of $a, b, c$ and $d$, i.e.

$$\max_{\mathbf{v} \in \{+1, -1\}^m} f(a, b, c, d) + \langle \mathbf{v}, t \rangle.$$

The three subset losses listed in main text take this form: $\mathbf{L}_{\text{IsErr}}(\mathbf{v}, y) = I(b = 0)I(c = 0)$, $\mathbf{L}_{\text{Ham}}(\mathbf{v}, y) = b + c$, and $\mathbf{L}_{\text{ErrSetSize}}(\mathbf{v}, y) = bc$. The key idea is that though the max itself is over $2^m$ quantities, there are only $O(m^2)$ possible values for $C = \begin{bmatrix} a & b \\ c & d \end{bmatrix}$ — note that fixing $a$ (where $0 \leq a \leq m$) also fixes $c = |y| - a$, and similarly fixing $d$ (where $0 \leq d \leq m$) also fixes $b = m - |y| - d$. For any fixed $C$ (i.e. fixing $a, b, c$ and $d$), let $\mathcal{V}_{abcd}$ denote the set of vectors $\mathbf{v}$ that yield $C$. We can compute $\max_{\mathbf{v} \in \mathcal{V}_C} \langle \mathbf{v}, t \rangle$ in closed form, because the objective to be maximized is linear. Let $\mathcal{I}_{pos}$ denote the classes in $y$ sorted in decreasing order of $t$. Let $\mathcal{I}_{neg}$ denote the classes *not* in $y$ sorted in decreasing order of $t$. Now, for $\mathbf{v}$ to be optimal, we set the values corresponding to the first $a$ indices in $\mathcal{I}_{pos}$ to 1, the remaining indices in $\mathcal{I}_{pos}$ to -1, the last $d$ indices in $\mathcal{I}_{neg}$ to $-1$, and the remaining indices in $\mathcal{I}_{neg}$ to $+1$.

The procedure can be implemented with two `for` loops, where in the innermost `for` loop, we will set $\mathbf{v}_C^*$ that maximizes $\max_{\mathbf{v} \in \mathcal{V}_C} \langle \mathbf{v}, t \rangle$, compute $J_C = f(a, b, c, d) + \langle \mathbf{v}_C^*, t \rangle$ and keep track of the best $J_C$ so far. Finally we note that faster implementations can be obtained for specific functions $f(a, b, c, d)$.

## D    Proofs for Results in Section 4

*Proof of Lemma 3.* Both the surrogate as well as its gradient (w.r.t. $t$) can be computed if we can compute

$$\tilde{\sigma}_{t,y} = \operatorname*{argmax}_{\sigma \in [k]} \left( \mathbf{L}(\sigma, y) - \left\langle \operatorname{pred}^{-1}(\sigma_y) - \operatorname{pred}^{-1}(\sigma), t \right\rangle \right)$$

Let $\mathbf{L}(\sigma, y)$ be a loss derived from an NDCG type gain function. That is, let

$$\mathbf{L}(\sigma, y) = 1 - \frac{1}{W(y)} \sum_{i=1}^{m} \frac{F(y(i))}{G(\sigma(i))}$$

for some monotonically increasing functions $F, G$ and

$$W(r) = \max_{\sigma} \sum_{i=1}^{m} \frac{F(y(i))}{G(\sigma(i))}.$$

Note that $W(r)$ can be computed easily by sorting $y$. Since $\mathrm{rep}(\sigma) = \mathrm{pred}^{-1}(\sigma) = f(\sigma)/Z$, where $Z = \sqrt{\sum_i f^2(i)}$, we have,

$$\begin{aligned}
\tilde{\sigma}_{t,y} &= \operatorname*{argmax}_{\sigma \in [k]} \; \left( \mathbf{L}(\sigma, y) - \left\langle \mathrm{pred}^{-1}(\sigma_y) - \mathrm{pred}^{-1}(\sigma), t \right\rangle \right) \\
&= \operatorname*{argmax}_{\sigma \in [k]} \; \left( \mathbf{L}(\sigma, y) + \left\langle \mathrm{pred}^{-1}(\sigma), t \right\rangle \right) \\
&= \operatorname*{argmax}_{\sigma \in [k]} \; \left( 1 - \frac{1}{W(y)} \sum_{i=1}^{m} \frac{F(y(i))}{G(\sigma(i))} + \frac{1}{Z} \sum_{i=1}^{m} f(\sigma(i)) t_i \right) \\
&= \operatorname*{argmax}_{\sigma \in [k]} \; \left( \sum_{i=1}^{m} \frac{-F(y(i))}{W(y)G(\sigma(i))} + \frac{f(\sigma(i)) t_i}{Z} \right).
\end{aligned}$$

This is a linear assignment problem where the cost $C(i, j)$ of assigning item $i$ to position $j$ is

$$C(i, j) = \frac{-F(y(i))}{W(y)G(j)} + \frac{f(j) t_i}{Z}$$

which can be solved, e.g., using the $O(m^3)$ time complexity Hungarian algorithm (also known as Munkres' algorithm). $\qquad \square$

*Proof of Lemma 4.* Note that $y$ is sorted in decreasing order with relevance grade changes at positions $i_1, \ldots, i_N$. That is, the entries of $y$ obey the following ordering:

$$y(1) = \ldots = y(i_1 - 1) > y(i_1) = \ldots = y(i_2 - 1) > y(i_2) = [\ldots] = y(i_N - 1) > y(i_N) = \ldots = y(m)$$

Define $N+1$ sets $G_j = \{i_j, \ldots, i_{j+1} - 1\}$ for $j \in \{0, \ldots, k\}$ where $i_0 = 1$ and $i_{N+1} = m+1$ to handle boundary cases. If $t$ has margin $\gamma$ on $y$, it has to be first of all compatible with $y$. So entries of $t$ in group $j - 1$ should be equal to each other and larger than the entries in group $j$:

$$t_{i_{j-1}} = \ldots = t_{i_j - 1} > t_{i_j} = \ldots = t_{i_{j+1} - 1}$$

for $j \in [N]$. Moreover, we should have

$$\min_{\sigma \in \Sigma_y} \langle f(\sigma), t \rangle \geq \max_{\sigma' \notin \Sigma_y} \langle f(\sigma), t \rangle + \gamma'$$

where $\gamma' = \gamma Z$ and $Z$ is the normalization needed so that $f(\sigma)/Z$ is a unit vector.

Note that $\sigma' \notin \Sigma_y$ means that there is at least one "bad" pair $(i, i')$ such that $y(i) > y(i')$ (and hence $t_i > t_{i'}$) and yet $\sigma'(i) > \sigma'(i')$. We now claim that the maximum on the RHS is achieved at a $\sigma' \notin \Sigma_y$ with *exactly one* such bad pair. This is because, if we swap a bad pair in $\sigma''$ to get a new $\sigma'$ then $\langle f(\sigma''), t \rangle < \langle f(\sigma'), t \rangle$. So we can eliminate all bad pairs but one. This keeps us outside of $\Sigma_y$ and increases $\langle f(\sigma'), t \rangle$. We further claim that if there is exactly one bad pair $(i, i')$ then $i, i'$ have to be in adjacent groups with one of them right next to a group boundary. This is because, otherwise, there will have to be bad pairs other than $(i, i')$.

Let the bad pair be in groups $j - 1$ and $j$, i.e. $i \in G_{j-1}, i' \in G_j$ and $\sigma(i) = i', \sigma(i') = i$. The margin condition then says that,

$$f(i) t_{i_j - 1} + f(i') t_{i_j} \geq f(i') t_{i_j - 1} + f(i) t_{i_j} + \gamma'$$

which means

$$t_{i_j - 1} - t_{i_j} \geq \max_{i \in G_{j-1}, i' \in G_j} \frac{\gamma'}{f(i) - f(i')}$$

Since $f$ is strictly decreasing, the worst case is when $i = i_j - 1, i' = i_j$. This proves the lemma. $\quad \square$

*Proof of Corollary 5.* The condition $\|X_\tau\|_{\mathrm{op}}$ arises because the form of $\nabla_\tau$ in the subset ranking setting implies $\|\nabla_\tau\|_2 \le 2\|X_\tau\|_{\mathrm{op}}$. For $\mathbf{L} = \mathbf{L}_{\mathrm{NDCG}}$, $C_{\mathbf{L}} = 1$ and $c_{\mathbf{L}}$ can be computed as follows. Let $y$ be sorted in decreasing order of its entries. Minimum non-zero loss occurs if the last two documents are relevant and irrelevant get ranked incorrectly (errors higher up in the ranking will only incur more loss). So the minimum possible non-zero loss for a given $y$ is

$$
\frac{\frac{2^{y(m-1)}-1}{\log_2(1+m-1)} + \frac{2^{y(m)}-1}{\log_2(1+m)} - \left(\frac{2^{y(m-1)}-1}{\log_2(1+m)} + \frac{2^{y(m)}-1}{\log_2(1+m-1)}\right)}{\sum_{i=1}^m \frac{2^{y(i)}-1}{\log_2(1+i)}} \ge \frac{\frac{1}{\log_2 m} - \frac{1}{\log_2(m+1)}}{(2^{Y_{\max}}-1)\sum_{i=1}^m \frac{1}{\log_2(1+i)}}
$$

$$
= \frac{\frac{\log_2(1+1/m)}{\log_2 m \cdot \log_2(m+1)}}{(2^{Y_{\max}}-1)\sum_{i=1}^m \frac{1}{\log_2(1+i)}}
$$

$$
\ge \frac{\frac{1}{2m \cdot \log_2^2(m+1)}}{(2^{Y_{\max}}-1)\cdot \frac{m}{\log_2 2}}
$$

$$
= \frac{1}{2(2^{Y_{\max}}-1)m^2 \log_2^2(m+1)}.
$$

Therefore, the bound in Theorem 1 becomes

$$
\frac{4C_{\mathbf{L}}^2 R^2}{c_{\mathbf{L}}\gamma^2} \le \frac{2^{Y_{\max}+3}m^2 \log_2^2(2m)R^2}{\gamma^2}.
$$

$\square$

*Proof of Corollary 6.* We use Theorem 15 with $\psi(w) = \|w\|_r^2$ where $r = \log p_0/(\log p_0 - 1)$. For such an $r$, $\|w\|_r \le \|w\|_1 \le 3\|w\|_r$. Also, $\|\cdot\| = \|\cdot\|_r$, $\|\cdot\|_\star = \|\cdot\|_q$ where $q = \log p_0$. Note that $\psi$ is $(r-1)$-strongly convex w.r.t. $\|\cdot\|_r$ (see, e.g., [17]). Since

$$
\|X^\top u\|_q \le 3\|X^\top u\|_\infty \le 3\|X^\top\|_{2\to\infty}\|u\|_2
$$

the norm $\|\|\cdot\|\|$ is simply $3\|X^\top\|_{2\to\infty}$ where $\|X^\top\|_{2\to\infty} = \max_{j=1,\dots,p}\|X_j\|_2$ ($X_j$'s are columns of $X$). Since $\|w^\star\|_r \le \|w^\star\|_1$ there obviously exists a unit $\ell_r$ norm vector that has margin at least $\gamma$ on the dataset. The bound in Theorem 15 then becomes

$$
\frac{4(3R)^2 C_{\mathbf{L}}^2}{(r-1)c_{\mathbf{L}}\gamma^2} \le \frac{36R^2 \log p_0 C_{\mathbf{L}}^2}{c_{\mathbf{L}}\gamma^2}.
$$

Corollary nows follows by using the bounds for $C_{\mathbf{L}}, c_{\mathbf{L}}$ from proof of Corollary 5. $\square$

# E   Proofs for Results in Section 5

*Proof of Corollary 7.* Corollary follows immediately from Theorem 1 and the $C_{\mathbf{L}}, c_{\mathbf{L}}$ calculations in the proof of Corollary 5. $\square$

*Proof of Corollary 8.* We use Theorem 15 with $\psi(w) = \|W\|_{2,r}^2$ where $r = \log p/(\log p - 1)$. For such an $r$, $\|W\|_{2,r} \le \|W\|_{2,1} \le 3\|W\|_{2,r}$. Also, $\|\cdot\| = \|\cdot\|_{2,r}$, $\|\cdot\|_\star = \|\cdot\|_{2,q}$ where $q = \log p$. Note that $\psi$ is $(r-1)$-strongly convex w.r.t. $\|\cdot\|_{2,r}$ (see, e.g., [17]). Since

$$
\|uv^\top\|_{2,q} \le 3\|uv^\top\|_{2,\infty} = 3\|u\|_2\|v\|_\infty
$$

the norm $\|\|\cdot\|\|$ is simply $3\|\cdot\|_\infty$. Since $\|W^\star\|_{2,r} \le \|W^\star\|_{2,1}$ there obviously exists a unit group $(2,r)$-norm matrix that has margin at least $\gamma$ on the dataset. The bound in Theorem 15 then becomes

$$
\frac{4(3R)^2 C_{\mathbf{L}}^2}{(r-1)c_{\mathbf{L}}\gamma^2} \le \frac{36R^2 \log p \, C_{\mathbf{L}}^2}{c_{\mathbf{L}}\gamma^2}.
$$

Corollary nows follows by using the bounds for $C_{\mathbf{L}}, c_{\mathbf{L}}$ from proof of Corollary 5. $\square$