[Reviews · NeurIPS 2015]

Submitted by Assigned_Reviewer_1

The paper proposes a general framework of online linear regression with an instance space X \subseteq R^p, a finite label space Y, a finite prediction space S, and a loss function L: S \times Y \to R_+. The following protocol is slightly different from, but equivalent to the one proposed in the paper, but I think it is easier for me to understand. For each trial t=1,..,n, the following happens for the learner: (1) predict weight vectors w_{t,s} in R^p for all s in S. (2) observe an instance x_t in X, and compute the inner products z_s = w_{t,s}^T x_t for all s in S. (3) let s_t = argmax_{s in S} z_s and output s_t in S. (4) observe a label y_t in Y and incur loss L(s_t, y_t).

In the paper, the weight vectors w_{t,s} are assumed to form a rank d matrix. More precisely, the matrix (w_{t,1} ... w_{t,|S|}) should be represented as a product W_t B for a p \times d matrix W_t and d \times |S| matrix B, where B is a fixed and known. So, the problem is essentially to predict matrices W_t. Note that in the notation of the paper, W_t is defined as the transpose and B = (rep(1), ..., rep(|S|)).

Then the paper proposes a Perceptron-like algorithm and gives a mistake bound if the instance-label sequence (x_1,y_1)...,(x_n,y_n) is linearly separable with a certain margin. The paper also shows that many problems fall in this framework and so the algorithm can apply, such as binary classification, multiclass classification, ordinal regression, and multilabel classification. Moreover, the paper applies the method to ranking problems, where the label and/or prediction spaces are combinatorial sets.

The framework proposed looks similar to but slightly extends the existing framework of structured prediction. Moreover, I think that the paper gives a new vista of low rank matrix prediction. On the negative side, the paper only considers the linearly separable cases. It would be nicer if the regret analysis is also given. I wonder why Winnow-type or more general gradient dcsent type algorithms are not considered. I also wonder why the label space Y is restricted to be finite.

Minor comments: I do not understand why the "compatibility" is required for defining a margin. In Line 8 of Algorithm 1, \sigma_y is not well defined because it is not unique. In the paper, the matrix B is fixed and given. What if this is not the case?
Summary: The paper gives a new insight into structured prediction, although it provides no regret analysis.

Submitted by Assigned_Reviewer_2

I'm satisfied wit the rebuttal.

In particular, the explanation to my technical question turned out to be pretty simple when they pointed it out.

++++++++++

The basic framework is quite natural, yet new as far as I know.

The main algorithm and its analysis are perhaps not very surprising, but there is a lot of insight and non-trivial technical work in applying it to the various specific settings.

The experiments, though limited, are adequate for the purpose of illustrating how the theory works.

The paper is generally quite well written, both in terms of English and mathematics.

I was sometimes a bit confused by the notation, for example use of capital vs. lower case X and W (for example Corollary 7) and whether X in Section 4 is treated as matrix or vector.

Some typos: l. 398 "vey", l. 414 "significantally".

On line 321, the last C_L is should be squared.

Finally, a technical question: I'm confused about the assumptions and notation in Lemma 4 and its proof.

At first sight, the margin condition on lines 293-294 does not say anything about value t_i when i is not of the form i_j or i_j-1.

Intuitively to me, in this margin condition t_{i_j-1} should be the largest t_i in group i_j\leq i\leq i_j-1, and similarly for t_{i_j}.

In the proof you say that compatibility actually implies that all t_i values in the same group are equal, in which case what I just said is of course irrelevant, but I can't see why this follows from compatibility.

If compatibility indeed implies the equalities on line 863, I must really be missing something.

Could you please explain this a little.
Summary: The paper presents a new general online prediction framework, and algorithm with convergence bounds, that can be applied to a large variety of settings such as ranking and multilabel classification. The theory provides new insight to the notion of margin in different learning tasks and often gives bounds similar to ones obtained earlier by more specialised algorithms.

Submitted by Assigned_Reviewer_3

The paper proposes a new general perceptron-like algorithm, Predtron, for learning problems with large prediction and label spaces possibly of combinatorial nature, where predictions and labels are not necessary the same objects. First, a general mistake bound is given for the case in which the data is separable with a margin. Then, the algorithm is instantiated for several specific learning problems, such as binary classification (with or without a reject option), multiclass classification, ordinal regression, multilabel classification, subset ranking and multilabel ranking. The paper ends with a computational experiment.

The paper is very nicely written, clear and easy to follow. The results look technically sound (although I did not check all the proof in the appendix). It is hard for me to assess the novelty of the proposed algorithm. The algorithm is a version of perceptron based on a generalization of hinge loss for binary classification. Such a generalization was already considered before (mainly in the context of structural SVMs), and is usually referred to as structural hinge loss. I think the contribution here is to define separate prediction and label spaces and explicitly use representation of predictions (which is usually hidden in the feature vector for SVMs).

As far as I can tell, the mistake bound for the algorithm is a novel, original result. The proof is actually quite simple and more or less follow standard techniques from the analysis of online gradient descent (and its generalization, online mirror descent, for Predtron with a link function) and from the mistake-bound analysis of the original perceptron algorithm.

What is a bit dissatisfying is that both the theoretical analysis and the experiment only concern a very special case of linear separability with margin. This is a very strong assumption on the data, and rarely appears in practice. For instance, the labels (relevance judgments) generated in the experiment are deterministic functions of the input X, so there is no label noise involved. It would be nice to see a counterpart of Theorem 1 in the nonseparable case, where the regret would be expressed as the total loss of the algorithm minus the total phi-loss (structural hinge loss) of the best W. A bound on such defined regret can be obtained for the standard perceptron algorithm (see e.g. Cesa-Bianchi, Lugosi: "Prediction Learning and Games", Chapter 12).

The results on subset ranking are clearly interesting, especially the analysis of representation function and the value of the margin (Lemma 4). However, from what I know, solving max within a surrogate for NDCG through a linear assignment problem is known; see, e.g.: Chapelle, Le, Smola: Large margin optimization of ranking measures Le, Smola, Chapelle, Teo: Optimization of Ranking Measures
Summary: Interesting and nicely written paper on a general perceptron-like algorithm for learning complex, combinatorial label spaces with performance guarantees (although the theoretical analysis and the experiment only apply to separable case, a bit unrealistic in practice).

Submitted by Assigned_Reviewer_4

The authors introduce a novel, unified framework that explicitly handles distinct label and prediction spaces. This enables the authors to subsume many common machine learning problems including binary and mulitclass classification, ordinal regression and multilabel classification into their framework. Further, the framework allows the incorporation of a variety of non-standard loss functions and non-Euclidean norms. They derive the Predtron, which is an extension of the classical Perceptron to this framework, and obtain a Perceptron-like tight mistake bound. The authors further extend this framework to subset ranking and multilabel ranking. Experimental results on synthetic data for the subset ranking problem demonstrates a proof-of-concept of the approach.

This is a cool framework. And while the authors do clearly set out their goals for their experiments: as a confirmation of the behavior predicted by the bound, I'm disappointed that this did not include a feature construction for a search query on a small corpus of documents.
Summary: The proposed framework is novel and abstract, and broadly includes many different and recently relevant machine learning tasks and problems. Experimental results are proof-of-concept only, approach comes with a perceptron-like theoretical bound.

Author Feedback
Author rebuttal: We thank the reviewers for encouraging and insightful comments. Below we respond to some concerns raised in the reviews. We appreciate comments to improve readability -- those will be incorporated in camera-ready version.

Only separable case is considered in the paper: This was done solely for simplicity of presentation. Our proof technique is based on OCO analysis and will immediately yield relative loss bounds (target loss of algorithm bounded by best surrogate loss over linear predictors) when separability does not hold. This claim is easily verified by looking at line 565 which gives a bound that does NOT require separability. In fact, separability is used only at the very end of the proof (line 567-580).

Winnow-type and more general gradient descent algorithms are not considered: In fact, we do consider extensions albeit in the supplementary material (Alg. 3 - Predtron.Link). The extension is what enables us to prove Corollary 6 and Corollary 8, results that can be interpreted as winnow-type guarantees under the assumption that there is a good sparse classifier.

Reviewer 2's technical question: Thanks for delving deep into the proofs. The compatibility condition does indeed imply the equalities on line 863. The key
here is the requirement that the set of predictions that achieve the max in the definition of pred be *exactly* the same as the set of predictions that incur zero loss. So if there are multiple elements in \Sigma_y then those very elements should form the argmax in Eq. (1). In the ranking case, this means
that if multiple documents share the same relevance score then a compatible score vector t should have equal entries for those documents to make sure that
there are multiple rankings sigma whose representations maximize inner product with t.